# *In Vitro* Skin Retention of Crisaborole after Topical Application

**DOI:** 10.3390/pharmaceutics12060491

**Published:** 2020-05-28

**Authors:** Adriana Fantini, Anna Demurtas, Sara Nicoli, Cristina Padula, Silvia Pescina, Patrizia Santi

**Affiliations:** Department of Food and Drug, University of Parma, Parco Area delle Scienze 27/a, 43124 Parma, Italy; adriana.fantini@unipr.it (A.F.); anna.demurtas@studenti.unipr.it (A.D.); sara.nicoli@unipr.it (S.N.); cristina.padula@unipr.it (C.P.); silvia.pescina@unipr.it (S.P.)

**Keywords:** atopic dermatitis, crisaborole, skin retention, tape stripping

## Abstract

Crisaborole, a nonsteroidal phosphodiesterase 4 inhibitor, represents the first nonsteroidal medication approved for the treatment of atopic dermatitis in over a decade. In this work, crisaborole skin permeation and retention was studied *in vitro* from a 2% ointment using porcine skin as barrier. Crisaborole was also characterized in terms of thermal behavior, solubility, and logP. Control experiments were performed also on tape stripped skin to clarify the role of stratum corneum in drug partitioning and permeation across the skin. The results obtained indicate that crisaborole accumulates into the skin in considerable amounts after application of a topical lipophilic ointment. Crisaborole shows more affinity for the dermis compared to the epidermis despite its relatively high value of partition coefficient; stratum corneum analysis revealed a low affinity of the drug for this skin layer. Skin penetration across hair follicles or sebaceous glands can be a reason for the high dermis retention and is worth further investigation. The comparison with data obtained from a solution in acetonitrile suggests that the formulation plays a certain role in determining the relative distribution of crisaborole in the skin layers and in the receptor compartment.

## 1. Introduction

Atopic dermatitis (AD), known also as atopic eczema, is a chronic inflammatory condition of the skin [1] which affects up to 5% of adult population and 15%–20% of children [2,3]. AD is a significant cause of morbidity, quality-of-life impairment, and health care costs, with a significant social and economic impact worldwide [4]. AD has a strong genetic component and increases the propensity towards the development of other allergic diseases, such as asthma, allergic rhinitis, and food allergy in later childhood, described as the “atopic march” [5].

AD has a broad clinical spectrum and a fluctuating course, with manifestations that vary with age; common characteristics are intense itch and recurrent eczematous lesions [1,2,6].

It is generally accepted that several mechanisms contribute to AD etiology and manifestations [1,6,7]: impairment of epidermal barrier function, deficiency in the structural protein filaggrin, dysbiosis of the skin microbiota (in particular increase of *Staphylococcus aureus*), systemic immune responses, and neuroinflammation, which is involved in itch.

The reference treatment for AD for many years has been topical corticosteroids. For more severely affected subjects, systemically administered immunosuppressants are often necessary [8,9], whereas topical calcineurin inhibitors (tacrolimus and pimecrolimus) entered the topical treatments for mild AD [10]. Despite their efficacy, the use of both steroids and calcineurin inhibitors is associated with safety concerns due to their side effects [11]. In fact, long-term topical steroid therapy produces skin atrophy and suppression of the hypothalamic–pituitary–adrenal axis, whereas the use of tacrolimus is associated with burning/stinging upon application.

Crisaborole is nonsteroidal phosphodiesterase 4 (PDE4) inhibitor and represents the first nonsteroidal medication approved for the treatment of atopic dermatitis in over a decade. PDE4 is a key regulator of inflammatory cytokine production in AD and its activity, is increased in circulating inflammatory cells of AD patients [11]. Clinical trials demonstrated its efficacy and safety [11], improving the quality of life of patients [12]. Crisaborole was approved in 2016 by the Food and Drug Administration for the treatment of AD in adult patients and was launched in 2017. The drug has been proposed also for the treatment of recalcitrant palmoplantar psoriasis [13] and, off-label, for psoriasis, seborrheic dermatitis, vitiligo, and inflammatory linear verrucous epidermal nevus [14].

The aim of this work was to study the delivery of crisaborole to the skin *in vitro*. Objectives of the study included (i) the characterization of crisaborole, in terms of thermal behavior, solubility, and logP; (ii) the determination of drug retention in the various skin compartments (stratum corneum, epidermis, and dermis) after application of a topical ointment; and (iii) the evaluation of the penetration across the skin to estimate the risk of systemic effects upon topical application. Control experiments were performed also on tape stripped skin to clarify the role of stratum corneum in drug partitioning and permeation across the skin.

The experiments were performed *in vitro* using pig ear skin as a well-accepted model of human skin [15,16].

## 2. Experimental Part

### 2.1. Materials

Crisaborole (m.w. 251.045 g/mol) was purchased from Hangzhou Dayangchem Co. (Hangzhou City, China); oleic acid was from Alfa Aesar (Karlsruhe, Germany); propylene glycol was from A.C.E.F. (Fiorenzuola d’Arda, Italy); and isopropyl myristate, and octanol were from Sigma-Aldrich S.r.l. (Milano, Italia). Geleol^®^ (mono- and diglycerides, United States National Formulary), petrolatum, white paraffin, and Transcutol^®^ HP (diethylene glycol monoethyl ether) were a gift from Gattefossé (Saint-Priest Cedex, France).

For HPLC analysis, pure water (PureLab^®^ Pulse, Elga Veolia, UK) was used. Acetonitrile (ACN) was of HPLC grade; all other reagents were of analytical grade.

### 2.2. Methods

#### 2.2.1. Thermal Analysis

Differential scanning calorimetry (DSC) was performed on an indium calibrated Mettler DSC 821e instrument (Mettler Toledo, Columbus, OH, USA) driven by STARe software (Mettler Toledo, Columbus, OH, USA). DSC traces were recorded by placing accurately weighed quantities (3–5 mg) of sample in an aluminium pan, which was sealed and doubly pierced. Scans were performed between 25 and 200 °C at 5 °C/min under a flow of dry nitrogen (100 mL/min). The thermogram of crisaborole is presented in Appendix A.

#### 2.2.2. Solubility Determination

An excess amount of Crisaborole was added to 1.0 mL of vehicle, left under magnetic stirring for 24 h at room temperature, and then centrifuged for 10 min at 13,000 rpm. The concentration of drug in the supernatant was determined by HPLC analysis after appropriate dilution. Experiments were performed in triplicate, using the following solvents: octanol, ethanol, propylene glycol, oleic acid, isopropyl myristate, and water.

#### 2.2.3. HPLC Analysis

For the quantitative determination of Crisaborole, HPLC analysis was performed according to Reference [17] using an HPLC-UV system (Infinity 1260, Agilent Technologies, Santa Clara, CA, USA) with a reverse phase C18 Atlantis column (3.9 × 150 mm, 3 μm, Waters, Milford, MA, USA). The mobile phase, pumped at 1 mL/min, was a mixture of TFA (tri-fluoro acetic acid) 0.05%:ACN (55:45, *v*/*v*). The temperature of the column was maintained at 35 °C. The injection volume was 100 μL, and absorbance was monitored at 250 nm. As reported in Reference [17], the limit of quantification of the analytical method is 0.03 μg/mL and the linearity range is from 0.06 to 6 μg/mL.

#### 2.2.4. Determination of Octanol/Water Partition Coefficient

Partition coefficient of crisaborole between octanol and distilled water was determined using the shake flask method, following the guidelines of the European Chemical Bureau. Briefly, before the partition coefficient determination, the two phases were mutually saturated by shaking overnight at the same temperature (20 ± 2 °C) and in the same ratio of the partitioning experiments (water:octanol 4:1). Crisaborole was dissolved in water-saturated organic solution at concentration of 200 μg/mL. Part of this solution (8 mL) was transferred to 15-mL centrifuge tubes and added to 2 mL of presaturated octanol. The tubes were shaken for one hour (this time interval was enough to guarantee the equilibrium, as demonstrated by preliminary experiments); then, the two phases were separated by centrifugation. Crisaborole was quantified in both phases: 10 μL of the organic phase was sampled and added to 1 mL of mobile phase and analysed by HPLC. The aqueous phase was carefully sampled and analysed directly by HPLC. The partition coefficient was calculated as the ratio of organic phase and water concentration of crisaborole at equilibrium. Experiments were performed in triplicate.

#### 2.2.5. Ointment Preparation and Characterization

The ointment used in this paper contains (% by weight): propylene glycol (9%), white paraffin (5%), petrolatum (77%), mono- and diglycerides NF (7%) (Geleol^®^), and crisaborole (2%) [18].

The ointment was prepared by heating all the excipients in a water bath at 70 °C and by then letting them cool down to 40 °C when crisaborole was added. Crisaborole was not completely dissolved in the ointment (Appendix A).

#### 2.2.6. Tissue Preparation

For permeation and retention experiments, porcine skin, coming from the outer part of the ear, was used. The skin was excised post-sacrifice from Landrace and Large White (age 10–11 months, weight 145–190 kg) female and male animals supplied from a local slaughterhouse (Macello Annoni Spa, Madonna dei Prati, I) within 3 h from animal death. After removal of subcutaneous fat, skin samples were frozen at −20 °C and used within 3 months. Just before the experiment, the skin was examined for any sign of damage.

To mimic the barrier impairment produced by AD, the skin was preliminarily tape-stripped 30 times to remove most of the stratum corneum (until the transepidermal water loss reached a value of 20 g/m^2^ h) and was then submitted to the permeation experiment, according to Reference [19].

#### 2.2.7. Skin Permeation and Retention

Franz-type vertical diffusion cells (Disa, Milan, I), with a diffusion area of 0.6 cm^2^, were used. The full thickness skin, thawed at room temperature, was mounted between the two halves of the cell, with the stratum corneum facing the donor compartment. The receptor compartment was filled with 4 mL of degassed saline solution, kept under magnetic stirring, while in the donor compartment, the ointment was applied in finite dose conditions (*approx*. 10 mg/cm^2^). The receptor compartment was immersed in a thermostatted bath set at 37 °C to guarantee a skin surface temperature of 32 °C. Crisaborole solubility in the receptor solution was 0.11 mg/mL, enough to guarantee sink conditions. At the end of the application time (4, 8, 16, or 24 h), the skin was dismounted from the cell and a sample of receptor solution was taken for further analysis. The skin was wiped 3 times with paper soaked in distilled water, 3 times with paper soaked in ethanol, and then tape stripped once to remove the excess formulation. This cleaning procedure was validated in control experiments in which the formulation was removed immediately after application, and the skin was submitted to extraction and analysis: no crisaborole was found in the samples.

For epidermis/dermis separation, the skin was cut in correspondence of the permeation area and the skin was heated at 50 °C with a hair dryer for 30 s; then, the epidermis was separated from the dermis by scraping with a scalpel [20]. Epidermis and dermis samples were placed in glass vials and extracted with 1 mL of TFA 0.05%:ACN (55:45, *v/v* for epidermis—30:70, *v/v* for dermis) overnight at room temperature [17]. The stability of Crisaborole in the conditions used for epidermis/dermis separation has been demonstrated previously [17].

In a set of experiments, the amount of drug accumulated in the stratum corneum was also determined: the skin was submitted to the tape stripping procedure [21] before epidermis/dermis separation. After removal of the preparation, the permeation area was defined using a polyethylene liner template fixed to the skin. The removal of stratum corneum was performed with 25 adhesive tape strips (Scotch^TM^ 845, 3M, Saint Paul, MN, USA) and weighed before and after the application on the skin in order to know the amount of stratum corneum removed. For each experiment, the tape strips were pooled in five sets (tape strips# 1–5, 6–10, 11–15, 16–20, and 21–25) in 5 test-tubes and extracted with 2.5 mL of TFA 0.05%:ACN (55:45, *v/v*) overnight at room temperature, according to a validated method [17]. The skin was then submitted to epidermis/dermis separation (see above).

The conditions tested and the number of replicates for each condition are reported in Section 3.2. 

#### 2.2.8. Data Analysis

All data are reported as mean value ± sd. Statistical differences were evaluated by one-way analysis of variance (one-way ANOVA) with Bonferroni post hoc (level of significance *p* < 0.05).

## 3. Results and Discussion

### 3.1. Crisaborole Characterization

Crisaborole is relatively new molecule, and little has been published on its physicochemical properties. For this reason, the drug was characterized for thermal behavior, partition coefficient, and solubility.

#### 3.1.1. Thermal Analysis

The literature reports the existence of different polymorphs of crisaborole [22,23]. For this reason, we performed a preliminary DSC analysis to identify the polymorph(s) present in the commercial sample.

The DSC trace of commercial crisaborole is reported in Appendix A and shows two main endothermic peaks: one at 136.5 °C (127.21 J/g) and another at 178.23 °C (28.41 J/g). The second peak (178.23 °C) can be easily attributed, according to Campillo-Alvarado et al. [22], to the heat-induced rearrangement of crystal structure and is present in all the DSC traces of the polymorphs described by these authors.

The first peak (136.5 °C) occurs at a higher temperature compared to the forms I (129.1 °C) and II (125.6, 131.8 °C) described by Reference [22] and may indicate the presence of a different polymorphic form. In fact, Campillo-Alvarado et al. [22] demonstrated the presence of different crisaborole forms in the raw material obtained from different suppliers; additionally, a recent patent [23] reports the isolation and charaterization of 4 different polymorphic forms, with endothermic peaks included between 125 and 141 °C, obtained in different crystallization conditions.

#### 3.1.2. Partition Coefficient

Octanol/water partition coefficient were determined using the shake-flask method. Experimental logP_oct/w_ resulted 2.78 ± 0.16, which matches quite well with the value predicted using ACD/Labs software (logD_7.4_ 2.83). The logP value, together with the relatively low molecular weight, makes crisaborole a good candidate for skin application. In fact, using the Potts and Guy equation [24], the predicted permeability coefficient across the skin results *approx*. 1 × 10^−6^ cm/s, i.e., 5 × 10^−3^ cm/h.

#### 3.1.3. Solubility

The solubility of crisaborole in some of the most common solvent used for topical application is reported in Table 1, together with the solubility parameter of the solvents.

Among the solvents tested, crisaborole shows the maximum solubility in octanol, followed by ethanol; the solubility in propylene glycol seems to be still quite high, whereas it shows very low solubility in very hydrophilic (water) and very hydrophobic (isopropyl myristate) solvents.

In the abovementioned patent [23], the authors prepared and characterized four polymorphs, one of which (form I) is freely soluble in ethanol and propylene glycol and may be the form used in the commercial ointment.

### 3.2. Crisaborole Skin Permeation and Retention from Ointment

Skin retention and skin permeation offer valuable information concerning the concentration that the active can reach in the skin tissue (the site of action) and the possible side effect (due to systemic exposure), respectively. The formulation tested is an ointment, based principally on petrolatum, similar to the commercial formulation Eucrisa^®^; in the formulation prepared, crisaborole was present in part in solid form (see Appendix A).

In this study, the formulation was applied for different times (4, 8, 16, and 24 h) on intact or damaged skin (to simulate the barrier impairment produced by AD); the concentration of the drug was determined in the epidermis and dermis and in the receptor compartment. In a set of experiments, stratum corneum distribution was studied as well. The conditions tested are reported in Table 2.

#### 3.2.1. Time Course of Crisaborole Skin Penetration and Retention

Figure 1 reports the permeation profile of crisaborole across the skin, whereas Figure 1b reports the amount of crisaborole recovered in the skin as a function of time.

The skin permeation profile of crisaborole is almost linear with time after an initial lag phase. Crisaborole flux, estimated as slope of the linear portion (8–24 h) of the permeation profile resulted in the order of 0.2 μg/cm^2^ h (this is only an estimate; the data reported in Figure 1a derive from different experiments so it is not correct to calculate the flux as the slope of the line). Because crisaborole is not completely solubilized in the ointment (see Appendix A) the permeability coefficient cannot be calculated. The total amount of crisaborole recovered in the skin follows a similar trend, although the amount recovered seems to level off after 4–8 h and remain constant up to 24 h. This suggests the achievement of steady state in the skin after 8 h.

Crisaborole retention in the epidermis and dermis (Figure 1c) increased up to 8 h of application time and then remained constant, suggesting that steady-state was reached. It is interesting to observe that the total amount of crisaborole recovered in the dermis was higher compared to the epidermis (Figure 1c). When the data were normalized by the weight of tissue (Figure 1d), drug concentration was higher in the epidermis in consideration of its lower weight.

In general, the variability of the data resulted quite high, apart from the data reported in Figure 1d, suggesting that the main source of variability is the skin and the variation of its thickness/weight in the different specimen.

Crisaborole is a relatively lipophilic molecule, as witnessed by the logP value (2.78), so it may form a reservoir in the stratum corneum; for this reason, tape stripping experiments were performed.

#### 3.2.2. Stratum Corneum Distribution

This set of experiments served to elucidate the distribution of Crisaborole inside the stratum corneum. Additionally, the knowledge of drug concentration in this layer enabled us to calculate the percent distribution of the drug in the various skin compartments, i.e., stratum corneum, viable epidermis, and dermis.

Figure 2 reports the amount of crisaborole recovered in stratum corneum tape strips pooled by 5 at two experimental times (4 h and 16 h). As expected, crisaborole was more concentrated in the outer stratum corneum layers and decreased exponentially with depth. The difference between the two application times was very small and generally not significant (the only statistical difference was found in the 11–15 strips pool), although the general tendency was a lower concentration after 16 h of application when steady-state was already reached (see data on skin permeation).

#### 3.2.3. Crisaborole Distribution in the Skin Compartments

Considering the data obtained in the tape stripping experiments at 4 and 16 h (Figure 2), the percentage distribution of the drug in the various skin compartments (stratum corneum, viable epidermis, dermis, and receptor compartment) was calculated and is reported in Figure 3a. It is very interesting to observe that the variability of the data was much reduced, indicating that the main source of variability is, once more, the biological difference of skin specimen. Additionally, the distribution data confirms that most of crisaborole is recovered in the dermis, both at 4 and 16 h of application time. At 16 h application time, the percentage permeated was significantly higher compared to 4 h (*p* < 0.05); the reverse situation was found in the percentage in the epidermis (the difference was not statistically significant).

It should be underlined that the epidermis/dermis separation after tape stripping is not always easy, so the percentage recovered in the dermis might be overestimated because it contains traces of epidermis. For this reason, the same data elaboration was performed on the data reported in Figure 1; the result is reported in Figure 3b. The data reported in this figure confirm the result obtained in the tape stripping experiments, i.e., crisaborole tends to accumulate in the dermis; application time has a modest effect and produces an increase in the percentage of drug recovered in the receptor compartment, although not statistically significant.

The preferential accumulation of crisaborole in the dermis may support a transappendageal transport across hair follicles [27].

Crisaborole distribution in epidermis, dermis, and permeated are compared to our previous data [17] obtained in similar experimental conditions from a solution in acetonitrile to evaluate the effect of formulation/excipient. These data, reported in Figure 3c, suggest that the formulation plays a certain role in determining the relative distribution of crisaborole in the skin layers and in the receptor compartment. When the drug was dissolved in ACN, *approx*. 50% of the total amount recovered was found in the epidermis, much more compared to the ointment (*p* < 0.05). In the experiments done with ACN, crisaborole was not extracted from the stratum corneum, so it is not known if the drug forms a reservoir in the stratum corneum when applied as ACN solution.

From the results obtained, it appears that the use of an ointment is superior compared to a solution in a volatile solvent because it allows more drug to be retained in the dermis, where the disease is located. The results obtained show also a very low retention in the stratum corneum despite the lipophilic nature of the drug: this might be due to the low solubility in the stratum corneum lipids, which is supported by the relatively low solubility in isopropyl myristate (a solvent often employed to simulate stratum corneum lipids). Another reason for the low stratum corneum retention can be also the penetration pathway, which is not forcedly diffusion across the skin layers but might be via skin appendages.

#### 3.2.4. The Role of Stratum Corneum

The barrier associated with the skin is normally presented by the stratum corneum. To assess the role of this permeation barrier, the skin was preliminarily tape-stripped to remove most of the stratum corneum and then submitted to the permeation experiment: these conditions have been shown to mimic the barrier impairment produced by AD [19].

The results obtained are reported in Figure 4. In the absence of stratum corneum, both skin retention and skin permeation increased to a considerable extent; the enhancement factor obtained in the epidermis was *approx.* 6, whereas it reached 10 in the dermis. When the stratum corneum was partly removed by tape stripping, crisaborole was found in significant amounts also in the receptor compartment (enhancement factor = 8).

From these results, it appears that the damage produced by AD to the skin may increase to a significant extent the concentration of crisaborole in all the skin compartments, particularly in the dermis.

## 4. Conclusions

The results obtained in the present work indicate that crisaborole accumulates into the skin in considerable amounts after application of a topical lipophilic ointment. Crisaborole shows more affinity for the dermis compared to the epidermis despite its relatively high value of partition coefficient; stratum corneum analysis revealed a low affinity of the drug for this skin layer. Skin penetration across hair follicles or sebaceous glands or great affinity for sebum can be a reason for the high dermis retention and is worth further investigation.

Considering the time course of crisaborole skin penetration and retention, it appears that steady state is reached after 4–8 h, confirming the good permeation properties of crisaborole.

Tape stripping experiments confirmed that crisaborole does not form a reservoir in the stratum corneum when applied as lipophilic ointment.

The comparison with data obtained from a solution in acetonitrile suggests that the formulation plays a certain role in determining the relative distribution of crisaborole in the skin layers and in the receptor compartment.

## Figures and Tables

**Figure 1 pharmaceutics-12-00491-f001:**
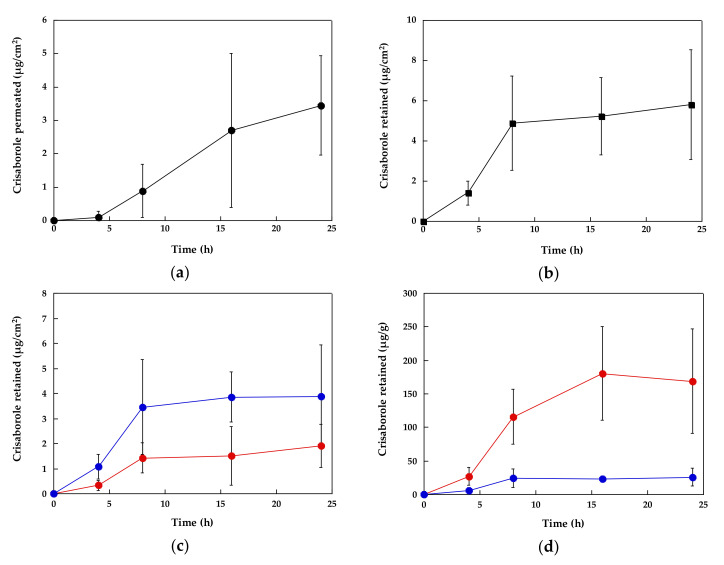
Crisaborole skin permeation (**a**) and total skin retention (**b**), and epidermis (**red line**) and dermis (**blue line**) retention expressed per unit surface (**c**) or per unit weight (**d**) after application of the ointment. Average values ± sd.

**Figure 2 pharmaceutics-12-00491-f002:**
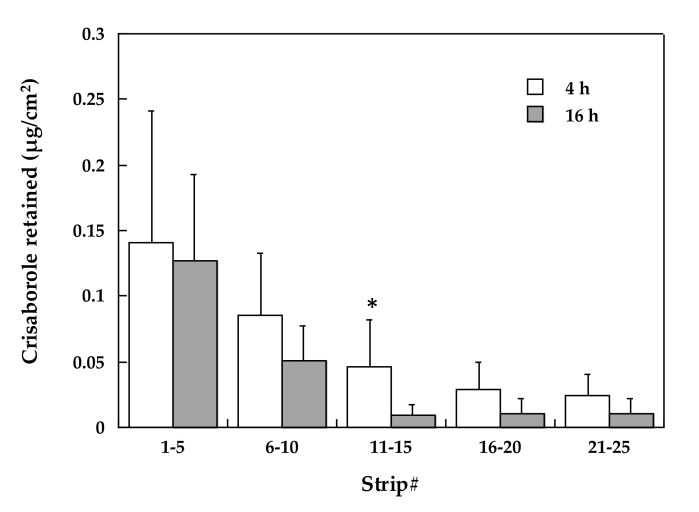
Recovery in the stratum corneum after application of the ointment for 4 (**white bars**) or 16 h (**grey bars**): Average values ± sd; * denotes statistical difference (*p* < 0.05).

**Figure 3 pharmaceutics-12-00491-f003:**
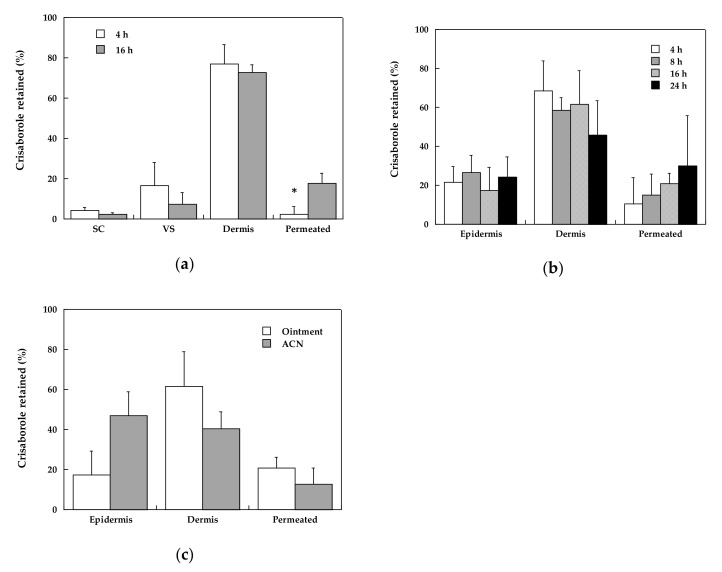
(**a**) The percentage distribution of crisaborole following the application of 2% ointment for 4 and 16 h; (**b**) the percentage distribution of crisaborole as a function of application time; and (**c**) the distribution of crisaborole in the skin and receptor compartment following application of ointment and 2% solution in ACN for 16 h (from Reference [17]). Average values ± sd. * denotes statistical differences (*p* < 0.05).

**Figure 4 pharmaceutics-12-00491-f004:**
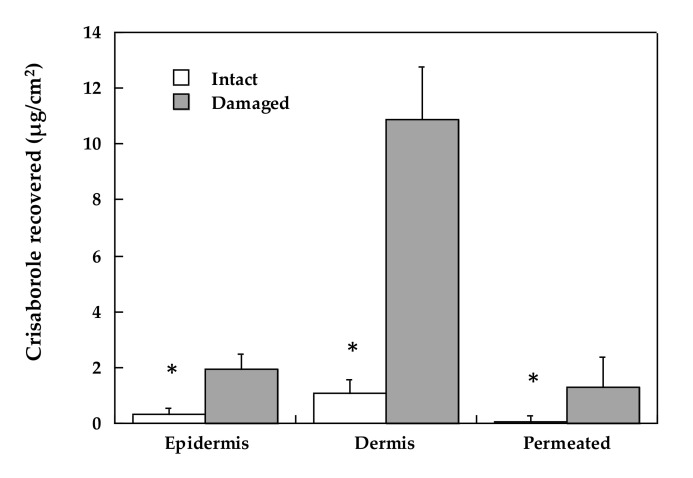
Tape stripping on crisaborole permeation and distribution in the epidermis and dermis after 4 h of ointment application. Average values ± sd. * denotes statistical differences (*p* < 0.05).

**Table 1 pharmaceutics-12-00491-t001:** Solvent properties and crisaborole solubility (average values ± sd).

Solvent	Solubility Parameter (MPa^1/2^)	Crisaborole Solubility (mg/mL)
Octanol	20.87 [25]	75.80 ± 1.70
Ethanol	26.52 [25]	40.6 ± 11.3
Propylene Glycol	30.22 [25]	12.20 ± 2.55
Oleic acid	17.39 [25]	3.44 ± 0.33
Isopropyl myristate	17.38 [26]	3.43 ± 1.68
Water	47.8 [25]	0.11 ± 0.01

**Table 2 pharmaceutics-12-00491-t002:** Conditions tested in skin permeation and retention experiments (the number of replicates is indicated in parentheses).

	2% Ointment	2% ACN Solution [17]
Intact Skin	Damaged Skin	Intact Skin
**Contact Time (h)**	**Epidermis/Dermis Retention**	**Stratum Corneum Distribution**	**Epidermis/Dermis Retention**	**Epidermis/Dermis Retention**
4	X (4)	X (4)	X (4)	
8	X (8)			
16	X (4)	X (4)		X (6)
24	X (7)

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
