# Peer review of "In Vitro Skin Retention of Crisaborole after Topical Application"

_pharmaceutics, 2020, doi:10.3390/pharmaceutics12060491_

Round 1

Reviewer 1 Report

The manuscript by Fantini et al. evaluated the Crisaborole permeation and retention in porcine skin in vitro using ointment formulation. The manuscript was clear written and fairly easy to follow. Some major concerns include:

  • How many porcine skins were used for section 3.2.1 in order to obtain the average and standard deviation for each data point? The standard deviations after 5 hr were very large and could indicate inappropriate experimental design.
  • The authors should perhaps explain further on Figure 2, especially in the methods of this corresponding section. It appears that the authors continued to “take-out” crisaborole from stratum corneum using a sample of 25, but grouped them each in 5. How was the grouping done? What was the purpose?
  • The authors should give basic observations and measurements of the porcine skin according to Section 2.2.5 as different skin samples could affect the permeation and retention scale.
  • The authors could consider coupling of fluorescent particles with crisaborole to further examine the histology of the porcine skin for permeation and retention.

Author Response

How many porcine skins were used for section 3.2.1 in order to obtain the average and standard deviation for each data point? The standard deviations after 5 hr were very large and could indicate inappropriate experimental design.

A: the number of replicates for each condition is now reported in Table 2. For each experiment, the skin coming from at least 3 different animals was used, to account for skin variability.

The authors should perhaps explain further on Figure 2, especially in the methods of this corresponding section. It appears that the authors continued to “take-out” crisaborole from stratum corneum using a sample of 25, but grouped them each in 5. How was the grouping done? What was the purpose?

A: the paragraph relative to tape stripping has been completely re-written. Tape stripping was performed to determine the profile of crisaborole in the stratum corneum. Tape strips pooling 5 by 5 is a compromise between detail of information and amount of analysis to be done. Additionally, by pooling the strips, we are sure to have samples with a concentration above the LOQ of the analytical method.

The authors should give basic observations and measurements of the porcine skin according to Section 2.2.5 as different skin samples could affect the permeation and retention scale.

A: the race and age of the animals has been added to the considered section. 

The authors could consider coupling of fluorescent particles with crisaborole to further examine the histology of the porcine skin for permeation and retention.

A: the suggestion of the reviewer is very interesting, but it is not feasible at this stage of the work. Furthermore, the conjugation of crisaborole with a fluorescent probe or fluorescent particles, might influenced in a non-predictable and non-controllable way its diffusion within tissue.

Reviewer 2 Report

Dear authors,

This manuscript is relevant and fits the scope of Pharmaceutics. However, it needs some improvements / clarifications. This review needs some work before publication. 

Is thermal analysis relevant for the results obtained in this manuscript?

Solubility studies: The authors should provide solubility studies in propylene glycol once the ointment contains this ingredient.

Did the authors perform solubility studies to select the receptor phase (Franz Cells)?

Lines 104-105: How can the authors state: “Crisaborole was not completely solubilized in  the ointment”. Microscopy analysis should be perform.

Lines 181 and 182: Please clarify this sentence.

What is the control composition?

Skin permeation results: what should be the best model to fit these results?

What is the rationale behind the following procedure: two experimental times (4 h and 16 h) ?

Why ACN? The authors should perform the same experiment with a suitable topical solution.

Skin Tissue: The authors should present ethical documentation

Author Response

Q: Is thermal analysis relevant for the results obtained in this manuscript?

Apparently, in the commercial formulation the drug is dissolved (https://www.tga.gov.au/sites/default/files/auspar-crisaborole-190814.pdf - Particle size was not considered to be a critical attribute as the drug substance is fully dissolved in propylene glycol in the drug product) whereas in our case undissolved drug was present

This has been added to the text and the thermogram has been included in the Supplementary material.

Q: Solubility studies: The authors should provide solubility studies in propylene glycol once the ointment contains this ingredient.

A: the solubility in propylene glycol is included in Table 1.

Q: Did the authors perform solubility studies to select the receptor phase (Franz Cells)?

A: The solubility of crisaborole in saline solution is similar to the solubility in water (0.11 mg/ml) and it is high enough to guarantee sink conditions (the maximum concentration measured in the receptor compartment was 3 µg/ml). This consideration has been added also to the text.

Q: Lines 104-105: How can the authors state: “Crisaborole was not completely solubilized in  the ointment”. Microscopy analysis should be perform.

A: the authors observed the ointment under a polarized-light microscope and the appearance was not homogenous, indicating the presence of undissolved crisaborole. A sentence has been added to the text and the picture to the Supplementary Material.

Q: Lines 181 and 182: Please clarify this sentence.

A: The sentence has been re-written to make it clearer.

Q: What is the control composition?

A: the authors did not find “control composition” in the text

Q: Skin permeation results: what should be the best model to fit these results?

A: The data reported in Figure 1A derive from different experiments, so the authors did not apply any mathematical model in this paper.

Q: What is the rationale behind the following procedure: two experimental times (4 h and 16 h) ?

A: different experimental times were tested, to characterize the time course of crisaborole skin retention: in certain conditions four different experimental times were used (4, 8, 16 and 24h), in some conditions only two experimental times were tested to limit the number of experiments.

Q: Why ACN? The authors should perform the same experiment with a suitable topical solution.

A: the authors refer to a published paper, the experiment with ACN was not performed in the present work.

Q: Skin Tissue: The authors should present ethical documentation

A: Skin samples were a gift of a local abattoir; the samples came from animals slaughtered for human nutrition.

Reviewer 3 Report

The manuscript presents a study on the in vitro skin delivery of crisaborole. When considering the crisaborole action on atopic dermatitis, a dermal delivery is desirable. The permeation and retention assay is the main aspect of the paper. Authors have a publication (cited as ref. 19) where extraction methods herein described were already validated. However, the “Skin permeation and retention” methods should be clearly presented. Many results presented have no supportive information on Methods section.

General:

- Temperature: use a space between the number and the unit (ºC). Please, do write ”ºC” instead of “º C”.

Materials:

- Why Eucrisa was not used in the study? Is crisaborole solubilized in the commercial form? The drug was not completely solubilized in the formulation you prepared. This may led to an underestimated analysis.

- When describing ingredient suppliers, please, homogenize: ingredient was purchased/a gift from supplier (city, country)

Methods;

- Ln 92: After the stop: “Part of this solution (8 ml) was transferred …”

- What is the diffusion area of the Franz cell?

- HPLC analysis should appear before 2.2.3

- Ln 96: “added of 1 ml of TFA 0.05%:CH3CN 55:45” replace by “added of 1 ml of mobile phase…”

- Ln127: the tape-stripping was performed after epidermis-dermis separation? If yes, explain why. If not, the tape-stripping procedure should appear before the skin thermal treatment.

- Ln128: Describe in greater detail the tape-stripping technique: tape origin, if using a weight to press the tape, during what time. This procedure should follow a validated protocol.

- Ln 139: “… according to a validated method [19] …”

Results:

- What is the pertinence of the thermal analysis and the solubility assay for the drug skin delivery assay? Please, discuss the results achieved.

- When using logPoct/w or logD7.4, use oct/w and 7.4 as subscript text to facilitate reading.

- Ln 174-175: “In this study the formulation was applied for different times (4, 8, 16, 24 h) on intact or damaged skin (to simulate the barrier impairment produced by AD)”. No information in the methods section regarding the damaged skin, and how the damage is provoked, is presented. As well, no information in the methods is given regarding the 2% ACN solution. Why using ACN when no information on drug solubility in ACN was given? It is because the data incorporated in this study (Fig. 3c) are from ref. 19?

As much as I understand, the skin damage technique appears in lines 255-256 (Results). This information should move to methods section. To assess the integrity of the skin, the resistance of the skin in each Franz cell is usually measured using an LCR meter. Did you perform any integrity assay?

- Figure 2: How do you explain a lower concentration after 16 h of application than after 4h?

- Figure 3a and 3b: both express % as a function of time. What is the difference between % recovered and Crisaborole retained (%)? Have you checked the mass balance?

- Ln 263: please, define EF (enhancing factor?)

- Figure 4 – Legend: TS; Damaged or Tape-stripped should be used. TS is used here for the first time without definition.

- Experimental values from this study can be used to provide scientific justification for using an ointment, as is the commercial form of the product. Is it possible to include such correlation in the discussion?

Author Response

General:

- Temperature: use a space between the number and the unit (ºC). Please, do write ”ºC” instead of “º C”.

A: the text has been changed according to the reviewer’s comment.

Materials:

Q: Why Eucrisa was not used in the study? Is crisaborole solubilized in the commercial form? The drug was not completely solubilized in the formulation you prepared. This may led to an underestimated analysis.

A: unfortunately, Eucrisa is not available in Italy and it was not possible to obtained it from the US.

Apparently, in the commercial formulation the drug is dissolved (https://www.tga.gov.au/sites/default/files/auspar-crisaborole-190814.pdf - Particle size was not considered to be a critical attribute as the drug substance is fully dissolved in propylene glycol in the drug product) whereas in our case undissolved drug was present.

- When describing ingredient suppliers, please, homogenize: ingredient was purchased/a gift from supplier (city, country)

A: the suggestion of the reviewer has been taken into account.

Methods;

- Ln 92: After the stop: “Part of this solution (8 ml) was transferred …”

A: the suggestion of the reviewer has been taken into account.

- What is the diffusion area of the Franz cell?

A: the diffusion area of the Franz cell is 0.6 cm2.

- HPLC analysis should appear before 2.2.3

A: the suggestion of the reviewer has been taken into account.

- Ln 96: “added of 1 ml of TFA 0.05%:CH3CN 55:45” replace by “added of 1 ml of mobile phase…”

A: the suggestion of the reviewer has been taken into account.

- Ln127: the tape-stripping was performed after epidermis-dermis separation? If yes, explain why. If not, the tape-stripping procedure should appear before the skin thermal treatment.

A: the paragraph has been completely re-written.

- Ln128: Describe in greater detail the tape-stripping technique: tape origin, if using a weight to press the tape, during what time. This procedure should follow a validated protocol.

A: the paragraph has been completely re-written.

- Ln 139: “… according to a validated method [19] …”

A: the suggestion of the reviewer has been taken into account.

Results:

- What is the pertinence of the thermal analysis and the solubility assay for the drug skin delivery assay? Please, discuss the results achieved.

Solubility assays were done again because little is known about crisaborole; additionally, knowing drug solubility in hydrophilic and lipophilic solvents can give information about its ability to cross the skin/accumulate in some skin layers.

The text has been modified to account for this observation.

- When using logPoct/w or logD7.4, use oct/w and 7.4 as subscript text to facilitate reading.

A: the suggestion of the reviewer has been taken into account.

- Ln 174-175: “In this study the formulation was applied for different times (4, 8, 16, 24 h) on intact or damaged skin (to simulate the barrier impairment produced by AD)”. No information in the methods section regarding the damaged skin, and how the damage is provoked, is presented. As well, no information in the methods is given regarding the 2% ACN solution. Why using ACN when no information on drug solubility in ACN was given? It is because the data incorporated in this study (Fig. 3c) are from ref. 19?

As much as I understand, the skin damage technique appears in lines 255-256 (Results). This information should move to methods section. To assess the integrity of the skin, the resistance of the skin in each Franz cell is usually measured using an LCR meter. Did you perform any integrity assay?

A: the suggestion of the reviewer has been taken into account and the information on skin damage (obtained by the tape stripping technique) is now included in the “Tissue preparation” paragraph.

The data relative to ACN are taken from reference 17 (former reference 19).

Tissue integrity was evaluated by visual inspection.

- Figure 2: How do you explain a lower concentration after 16 h of application than after 4h?

A: the concentration at 16h was lower than at 4h, although the difference was significant only for one point. The reason for this result is not known: at 4h the steady state was not reached in the skin, whereas it was reached at 16h.

- Figure 3a and 3b: both express % as a function of time. What is the difference between % recovered and Crisaborole retained (%)? Have you checked the mass balance?

A: it was typo, the Y axis of figure 3a and 3b have the same meaning: figure 3a refers to experiments in which the drug was separately quantified in the stratum corneum, whereas figure 3b to experiments without tape stripping at 4 different application times.

No mass balance check was performed in these experiments: however, the mass balance was checked during the validation of the extraction procedure (cfr. Reference 17) and 95% of the drug applied was recovered.

- Ln 263: please, define EF (enhancing factor?)

A: enhancement factor was written instead of EF

- Figure 4 – Legend: TS; Damaged or Tape-stripped should be used. TS is used here for the first time without definition.

A: “damaged” is now used instead of “TS”

- Experimental values from this study can be used to provide scientific justification for using an ointment, as is the commercial form of the product. Is it possible to include such correlation in the discussion?

A: a sentence has been added to the discussion

Reviewer 4 Report

Had the ointment the same composition as the marketed formulation? Why do you prepare it instead of purchasing reference product?

Line 115. Aqueous buffer was used, sink conditions should be demonstrated if you want a proper evaluation of the permeated amounts in receptor medium

Line 125. Briefly describe the conditions of the separation technique. Stability of API under heat should be demonstrated.

Section 2.2.6. did you study the recovery of the extraction method? The recovery of biological samples is usually below 100% and according bioanalytical methods, this fact should be studied. Information about tape used in stripping experiment is required.

Section 2.2.7. indicate the LOQ and range of the analytical method

Section 3.1.1 please provide the thermogram

Line 159: use superscript

Table 1. indicate the meaning of the abbreviations

Line 173. You said API is in part, in solid form, but in line 104 you state that is totally solved. Please clarify or show evidence about API solubilization state in vehicle. This fact has impact on permeation profile.

Line 175. How skin is damaged should appear in material and method and the method employed to damage the skin.

Table 2. Why did you choose ACN as vehicle? Did is a low biocompatible solvent, ethanol is usually preferred. This is also no reported in material and method section. Indicate the number of replicates carried out for each condition.

Figure 1C and 1D. The conclusion showed in figures is different, in 1C the retention is higher in dermis and 1D the opposite. Which one represents the reality, or it is more appropriate? This should be discussed.

Figure 3. How are the percentages obtained? They are obtained considering the total administrated dose in the donor compartment (like a mass balance experiment, 100%= dose in donor compartment) or considering the total amount obtained after skin and receptor medium analysis? (100% = (SC) + E + D + RM). Generally, it is considered more appropriate the 1st approximation, because mass balance shows the experiment suitability (this involved the analysis of non-permeated drug). The total mass recovered (non-permeated + SC + E+ D + RM) after the experiment should be within the 90-110% of the dose administrated at the beginning of the experiment. Please consider that topical absorption is usually below 1% of the dose administrated. Which is in your case?

Conclusions. Authors are considering follicular targeting as permeation route, but this was not demonstrated in the articles. Conclusions should refer to your results. If you consider follicular targeting as a hypothesis this should appear in discussion section.

Author Response

Had the ointment the same composition as the marketed formulation? Why do you prepare it instead of purchasing reference product?

A: unfortunately, Eucrisa is not available in Italy and it was not possible to obtained it from the US; the authors decided to use an ointment.

Line 115. Aqueous buffer was used, sink conditions should be demonstrated if you want a proper evaluation of the permeated amounts in receptor medium

A: The solubility of crisaborole in saline solution is similar to the solubility in water (0.11 mg/ml) and it is high enough to guarantee sink conditions (the maximum concentration measured in the receptor compartment was 3 µg/ml). This consideration has been added also to the text.

Line 125. Briefly describe the conditions of the separation technique. Stability of API under heat should be demonstrated.

A: the conditions of epidermis/dermis separation have been indicated, as well as the refence to Crisaborole stability in the conditions used.

Section 2.2.6. did you study the recovery of the extraction method? The recovery of biological samples is usually below 100% and according bioanalytical methods, this fact should be studied. Information about tape used in stripping experiment is required.

A: the recovery of the extraction method has been published separately (cfr. Reference 17)

Section 2.2.7. indicate the LOQ and range of the analytical method

A: the LOQ and linearity range of the analytical have been included.

Section 3.1.1 please provide the thermogram

A: the thermogram has been added in the Supplementary material

Line 159: use superscript

A: the suggestion of the reviewer has been taken into account.

Table 1. indicate the meaning of the abbreviations

A: the suggestion of the reviewer has been taken into account.

Line 173. You said API is in part, in solid form, but in line 104 you state that is totally solved. Please clarify or show evidence about API solubilization state in vehicle. This fact has impact on permeation profile.

A: the text has been clarified and a picture of the image of ointment under polarized light microscope is included in the Supplementary material.

Line 175. How skin is damaged should appear in material and method and the method employed to damage the skin.

A: the suggestion of the reviewer has been taken into account and the information on skin damage (obtained by the tape stripping technique) is now included in the “Tissue preparation” paragraph.

Table 2. Why did you choose ACN as vehicle? Did is a low biocompatible solvent, ethanol is usually preferred. This is also no reported in material and method section. Indicate the number of replicates carried out for each condition.

A: the authors refer to a published paper, the experiment with ACN was not performed in the present work. The number of replicates for each condition is reported in Table 2.

Figure 1C and 1D. The conclusion showed in figures is different, in 1C the retention is higher in dermis and 1D the opposite. Which one represents the reality, or it is more appropriate? This should be discussed.

A: Figure 1c reports the absolute amount of Crisaborole recovered in the tissue (µg/cm2); Figure 1d reports the amount recovered divided by the weight of tissue, to give an indication of the concentration that the drug reaches in the tissue. Both data are relevant and informative, they simply give a different information.

Figure 3. How are the percentages obtained? They are obtained considering the total administrated dose in the donor compartment (like a mass balance experiment, 100%= dose in donor compartment) or considering the total amount obtained after skin and receptor medium analysis? (100% = (SC) + E + D + RM). Generally, it is considered more appropriate the 1st approximation, because mass balance shows the experiment suitability (this involved the analysis of non-permeated drug). The total mass recovered (non-permeated + SC + E+ D + RM) after the experiment should be within the 90-110% of the dose administrated at the beginning of the experiment. Please consider that topical absorption is usually below 1% of the dose administrated. Which is in your case?

A: the percentages obtained are calculated on the total amount of crisaborole recovered. Mass balance was not performed systematically in these experiments: however, the mass balance was checked during the validation of the extraction procedure (cfr. Reference 17) and 95% of the drug applied was recovered. In the experiments of reference 17, the percentage of Crisaborole recovered in the skin, with respect to the amount applied, resulted approx. 30% after 16h.

Conclusions. Authors are considering follicular targeting as permeation route, but this was not demonstrated in the articles. Conclusions should refer to your results. If you consider follicular targeting as a hypothesis this should appear in discussion section.

A: the part relative to follicular penetration/targeting has been removed from the Conclusion and is present only in the discussion.

Round 2

Reviewer 1 Report

The authors have addressed the comments from the suggestion, and thus, there is no more comments for the revised manuscript.

Reviewer 2 Report

The authors have performed significant improvement, so this paper can be published in the present form.

Reviewer 3 Report

The paper was improved according to comments with respect to the previous version.

Reviewer 4 Report

Work is correct for publication as presented